# Pareto Optimal Decisions in Multi-Criteria Decision Making Explained with Construction Cost Cases

Hubert Anysz [1,*], Aleksander Nicał [1], Željko Stević [2], Michał Grzegorzewski [1] and Karol Sikora [3]

1   Faculty of Civil Engineering, Warsaw University of Technology, 00-637 Warsaw, Poland;
    a.nical@il.pw.edu.pl (A.N.); michal.grzegorzewski.stud@pw.edu.pl (M.G.)
2   Faculty of Transport and Traffic Engineering, University of East Sarajevo,
    74000 Doboj, Bosnia and Hercegovina; zeljko.stevic@sf.ues.rs.ba
3   Faculty of Engineering and Information Sciences, University of Wollongong in Dubai,
    Dubai Knowledge Park, Dubai, UAE; KarolSikora@uowdubai.ac.ae
*   Correspondence: h.anysz@il.pw.edu.pl

**Abstract:** In multi-criteria decision-making (MCDM) problems the decision-maker is often forced to accept a not ideal solution. If the ideal choice exists, it would be certainly chosen. The acceptance of a non- ideal solution leads to some inadequate properties in the chosen solution. MCDM methods help the decision-maker to structure his needs considering different units, in which the properties of the solutions are expressed. Secondly, with MCDM tools the assessment of the available solutions can be calculated with consideration of the decision-maker's needs. The incorporation of the cost criterion into the decision maker's preferences calculation, and the solution assessment calculation, deprives the decision-maker of the ability to calculate the financial result of the decision he must make. A new multi-criteria decision making with cost criterion analysed at the final stage (MCDM-CCAF) method is developed based on principle of Pareto optimal decisions. It is proposed to exclude the cost criterion from the MCDM analysis and consider it at the final phase of the decision-making process. It is illustrated by example solutions with consideration of cost criterion and without it. It is proposed to apply the invented post-processing method to all MCDM analyses where the cost criterion of analysed variants is considered.

**Keywords:** multi-criteria decision making; MCDM; AHP; TOPSIS; FUCOM; MARCOS; cost criterion; construction cost; life cycle cost; LCC; MCDM-CCAF

## 1. Introduction

Multi-criteria decision making (MCDM) methods were invented for the cases where the choice of a solution has to be made and the solution has several properties inseparably connected to a certain solution. If it is possible to create the ideal solution, its components could be easily chosen and then incorporated. There are a lot of examples also in the construction industry for the choices where the solution can be only treated as a system with several inseparable properties. Cases where the set of properties match exactly the preferences of the decision-maker are rare. This lack of ideal matching of the properties and preferences means that if the choice is made—because it has to be made—some features are accepted in spite of the fact they are not of the highest level (defined by the needs, preferences of the decision-maker). The MCDM problem is illustrated in Figure 1. The necessity of giving-up any value of a certain feature implies the comparisons to another solution, where another feature is not on the highest level. Is it reasonable, e.g., to give up 6 dB of sound insulation in favor of a 10% increase of thermal insulation of an external wall? It depends on the decision-maker's preferences. It is not a rare case when even the decision-maker does not realize his preferences. Assessing them is a difficult task when there are several properties. It is even more difficult when the prices of analysed solutions are considered. The price is given for a solution, not separately for its properties. That

is why the MCDM methods are invented. A significant part of the methods is invented in the previous century e.g., [1–3], but their wide application and—sometimes—their imperfections push the scientist to invent the new MCDM methods or modify existing ones e.g., [4–8].

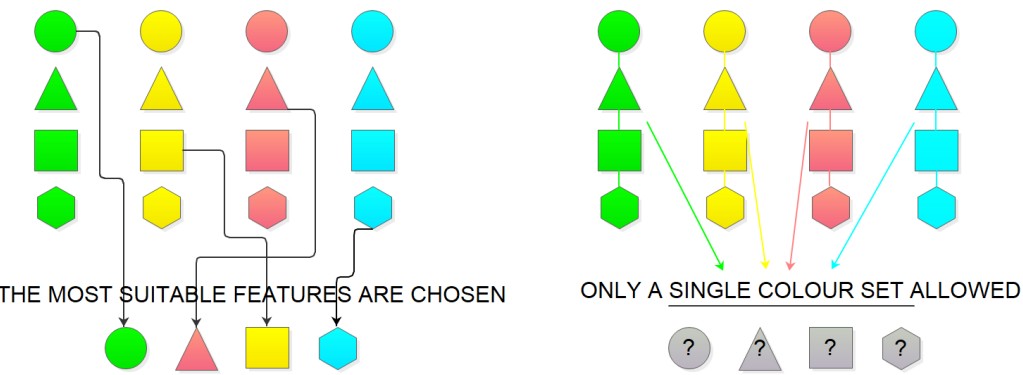

**Figure 1.** Schemas of MCDM problem (colour represents a solution, shape represents features).

Table 1 summarizes state-of-the-art on the variety of MCDM methods (concerning mainly the problems met in the construction industry and related ones as transport, logistics). They are also called MODM (multi-objective decision making as in [9]) or MADM (multi-attribute decision making as in [10,11].

**Table 1.** Comparative assessment in the AHP method.

| MCDM Method | Reference |
|:---:|:---:|
| AHP | [12–29] |
| ANP | [12] |
| ARAS | [10,15,30] |
| COPRAS | [30] |
| DELPHI | [12] |
| DEMATEL | [5,12,31–34] |
| EDAS | [30,33–36] |
| ELECTRE | [12,24,26,37] |
| ENTROPHY | [20,38] |
| FUCOM | [7,39,40] |
| MARCOS | [41] |
| MOORA | [9,10,30,42,43] |
| PROMETHEE | [14,18,24,30,44,45] |
| SAW | [18,22,44,46–49] |
| SWARA | [42] |
| TOPSIS | [14,16,24,26,27,30,43,44,46,47,50] |
| VIKOR | [51] |
| WASPAS | [39,42] |

The fuzzy sets theory invented by [52] is widely applied in solving MCDM problems (as e.g., in [53–59]). The feature of transforming imprecise verbal expressions of assessment, into crisp numbers, which can be ordered, made the fuzzy sets theory popular in MCDM applications. It was used in 18 analysed articles. The MCDM analysis becomes more complex when uncertainty issues are considered [4,60]. The construction industry is a specific one. Huge amounts of capital are usually spent on a construction project and almost every structure is unique, even if the technologies and materials applied in the building processes are repeatable (except for materials which are prepared on construction sites [61,62]). Structures are erected in different places with different earth conditions, and once erected the serve then under different climate conditions. Therefore, a solution suitable for one structure can be completely unsuitable for the similar structure erected

in the other place. Considering the aforementioned reasons, the decisions during for the construction process are of the highest importance, as their consequences are costly and long-lasting. To help the builders decide, MCDM methods are involved in working out the best decisions, i.e., decisions matching the best specific criteria applied to unique structures. Several examples can be found where multi-criteria decision-making serves for almost every stage of the construction processes. The application in the construction industry issues can be grouped and listed as for:

- planning purposes [9,17,53,63]
- a party selection (supplier, subcontractor, etc.) [22,33,36,41,54,57,64]
- construction material selection [42,45,56]
- construction technology selection [12,20,38,50,65]
- machinery selection [15,47]
- maintenance of the structures [66,67]
- sustainability issues [16,25,33,46,59,68]
- logistic problems [5,35,37,69]
- life cycle cost issues [46,50,70,71]
- transport problems [23,29,30,39,72]
- occupational risk issues [13,31]
- project assessments [11,43,51]
- time and cost issues [32,73,74]

The list of the applications of MCDM methods is not limited to the abovementioned subjects related to the construction industry. It should be emphasised that 37 analysed articles—almost 50% of articles illustrated with cases—use the cost as a criterion in MCDM. The importance of a single criterion and the problem of finding the set of criteria' weights, reflecting the best preferences of the decision-maker are analysed as a core issue e.g., in [7,60,75–78]. Examples of sensitivity analysis in MCDM can also be found, e.g., in [79].

The article aims to verify whether excluding the cost criterion from the MCDM analysis changes the result of the analysis. Moreover, the ranking of variants can be analysed at the end of MCDM analysis with consideration of the cost of each variant. This allows for a more conscious decision that considers the effectiveness issues of the variants to be chosen. It is achieved by selecting four MCDM methods (described in Section 2.1) and applying them for different cases (presented in Section 2.2). The results—rankings—are presented in Section 3. Section 4 contains the discussion of the results obtained for the cases when the cost criterion was included in MCDM methods, as well as, results where the cost criterion is excluded from MCDM analysis, but it is analysed jointly with rankings from MCDM methods. The structured, 6-stage method of analysis is proposed there. The term cost is often exchanged with the term price in the subsequent sections, as the price paid for the material or the system creates cost for the buyer (the decision-maker).

## 2. Materials and Methods

### 2.1. Description of Selected Methods

One of the most difficult problems in construction is to make objective decisions, especially for the selection of technology and material solutions [21]. Due to the complex nature of construction projects, the decision-making process can be extremely complicated and time-consuming. Not without significance are also subjective decisions taken by experts in the construction industry field. Difficulties often arise already at the stage of criteria set creation. A set of proper criteria and mathematical tools (such as computer calculation algorithms with multi-criteria analysis) could significantly improve objective decision-making [26]. Moreover, these mathematical tools can be implemented, as support in tendering procedures e.g., governmental megaprojects [80]. The paper presents the following multicriteria assessment methods (two well known, and two recently invented): Analytic Hierarchy Process (AHP), Technique for Order of Preference by Similarity to Ideal Solution (TOPSIS) [1], the FUCOM method [7,40], and the Measurement of Alternatives and Ranking according to a COmpromise Solution (MARCOS) method [41].

### 2.1.1. TOPSIS

The Technique for Order of Preference by Similarity to Ideal Solution (TOPSIS) algorithm was presented by the scientists Hwang and Yoon in 1981 [1]. In that paper, the classic model of the TOPSIS algorithm [1,81] is applied to carry out the multicriteria assessment. Before the explanation of the procedure, the following terms need to be defined:

$m$—the number of available solutions,
$n$—number of considered criteria,
$x_{ij}$—value of the $i$-th variant according to $j$-th criterion,
$X$—data matrix $[x_{ij}]$,
$Q$—vector of criteria weights $[q_1, q_2, \ldots, q_n]$.

In Step I the considered criteria are normalized if the values are impossible to be compared with each other. For this purpose, the Euclidean normalization can be applied, according to the formula:

$$z_{ij} = \frac{x_{ij}}{\sqrt{\sum_{i=1}^{n}(x_{ij})^2}} \tag{1}$$

In Step II the normalized decision matrix can be created, according to the following formula:

$$V = [q_j z_{ij}] \tag{2}$$

The following Step III includes the definition of reference vectors, i.e.,

- Positive Ideal Solution (PIS), marked as $A^+$, defined as:

$$A^+ = \{v_1^+, v_2^+, \ldots, v_n^+\} \tag{3}$$

where:

$$v_j^+ = \begin{cases} max_i v_{ij}, & j \in I \\ min_i v_{ij}, & j \in J \end{cases} \tag{4}$$

- Negative Ideal Solution (NIS), marked as $A^-$, defined as:

$$A^- = \{v_1^-, v_2^-, \ldots, v_n^-\} \tag{5}$$

where:

$$v_j^- = \begin{cases} max_i v_{ij}, & j \in J \\ min_i v_{ij}, & j \in I \end{cases} \tag{6}$$

In Step IV it is necessary to determine the distance of the available solutions from the reference vectors, according to the following formulas:

$$d_i^+ = \left( \sum_{j=1}^{n} (v_{ij} - v_j^+)^p \right)^{\frac{1}{p}} for \ i = 1, \ 2, \ \ldots, \ m \ and \ p = 2 \tag{7}$$

and:

$$d_i^- = \left( \sum_{j=1}^{n} (v_{ij} - v_j^-)^p \right)^{\frac{1}{p}} for \ i = 1, \ 2, \ \ldots, \ m \ and \ p = 2 \tag{8}$$

The formula for the synthetic measure $S$ used for the final evaluation of the solution with the TOPSIS algorithm is as follows [27]:

$$S_i = \frac{d_i^-}{d_i^- - d_i^+} for \ i = 1, \ 2, \ \ldots, \ m \ and \ S_i \in \langle 0, 1 \rangle. \tag{9}$$

The higher the value of $S$ for a given solution, the better the variant is.

### 2.1.2. AHP

The AHP is a four-step method with the following steps [2,3,82]:

- Step I—hierarchy of the problem
- Step II—definition of preferences by the decision-maker
- Step III—preference matrix consistency testing
- Step IV—creating a summary ranking

A diagram showing the first step of the AHP assessment is presented below as Figure 2 [27,83].

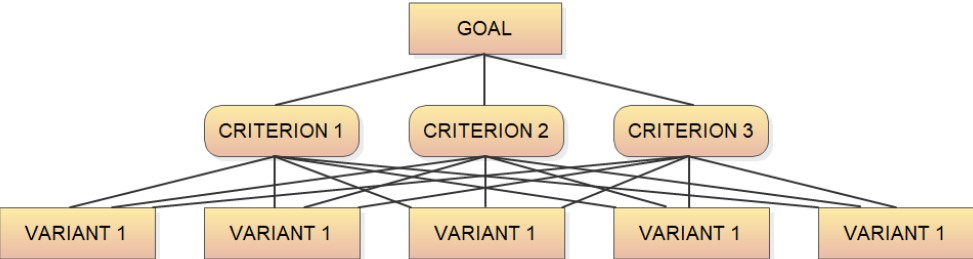

**Figure 2.** The structure of the hierarchy problem in the AHP method.

In Step II defining the preferences of the decision-maker using numerical values from 1 to 9 (less often from 1 to 7) is presented. Table 2 shows the values of the comparative assessment against each other. Values not listed in Table 2 (2, 4, 6, 8) characterize intermediate values. These values can be applied when the numerical interpolation of the criteria comparison is needed because the decision-maker cannot find the right words to describe the relationship between the given criteria [2,29].

**Table 2.** Comparative assessment in the AHP method.

| Comparative, Pairwise Assessment of A against B | Value |
|:---:|:---:|
| Just as good or important | 1 |
| A little better or more important | 3 |
| Definitely better or more important | 5 |
| Much better or more important | 7 |
| Extremely better or more important | 9 |

Preferences are specified for each level within the defined hierarchical structure. Only objects that are at one level of the hierarchy can be assessed against each other. The comparative assessment is subjective and is made by the decision-maker [27]. The result of Step II is a square matrix A in which the terms $a_{ij}$ illustrate the preferences of the decision-maker. The digits 1 are on the diagonal of the matrix A, there is also the reciprocal of the adopted preferences, i.e.,

$$a_{ij} = \frac{1}{a_{ji}}, \tag{10}$$

The next sub-step is to normalize matrix A to matrix B using the following dependence:

$$b_{ij} = \frac{a_{ij}}{\sum_{i=1}^{n} a_{ij}}, \tag{11}$$

The value $b_{ij}$ is expressed as the quotient of the term $a_{ij}$ to the sum of the terms in the *j*-th column of the matrix A. Weights of the examined elements ($w_i$) are the arithmetic means of the rows of the matrix B according to the following formula [2,28,83]:

$$w_i = \frac{1}{n} \sum_{j=1}^{n} b_{ij} \tag{12}$$

In Step III the decision-maker only assesses one pair of criteria each time with the following control coefficients introduced: Consistency Index ($CI$) and Consistency Ratio ($CR$)

$$CI = \frac{\lambda_{max} - n}{(n-1)}, \tag{13}$$

$$CR = \frac{\lambda_{max} - n}{RI \cdot (n-1)}, \tag{14}$$

where:

$\lambda_{max}$—the maximum eigenvalue of the matrix,
$RI$—the value of the average random consistency index $CI$ according to Table 3 presented below

**Table 3.** The values of the average random index of RI.

| Matrix Dimension n | 2 | 3 | 4 | 5 | 6 | 7 | 8 | 9 | 10 |
|---|---|---|---|---|---|---|---|---|---|
| $RI$ | 0 | 0.52 | 0.89 | 1.11 | 1.25 | 1.35 | 1.40 | 1.45 | 1.49 |

If $CR \leq 0.10$, then the preference matrix can be considered consistent. When $CR > 0.150$, the assumptions from step II should be changed [2,3].

The last step IV consists of creating a ranking of the available solutions in terms of their suitability to meet the main goal, in order from the best to the worst. The total score of a single variant can be calculated according to the following formula:

$$P = \sum_{i=1}^{n} w_i \cdot k_i \tag{15}$$

where:

$P$—final score for a given solution variant,
$w_i$—criterion weight according to the Formula (12),
$k_i$—evaluation of a given criterion.

### 2.1.3. FUCOM

The FUCOM method serves for the calculation of the weight of criteria and uses the comparison of paired criteria and the validation of results by deviating from the maximum consistency [7]. The main purpose of this method is a reduction of the subjectivity in the MCDM processes. Some of the advantages of this method are reduction of the number of pairs for comparison, consistency in comparing criteria, and contribution to rational judgment [8]. The FUCOM method consists of the following steps:

Step 1. Ranking of criteria/sub-criteria using expert preferences $C_{j(1)} > C_{j(2)} > \ldots > C_{j(n)}$.

Step 2. Calculation of the vector of the comparative significance of the evaluation criteria

$$\Phi = \left( \varphi_{1/2}, \varphi_{2/3}, \ldots, \varphi_{k/(k+1)} \right) \tag{16}$$

Step 3. Defining the constraints of a nonlinear optimization model. The values of the weighting coefficients should satisfy two conditions:

Condition 1. The ratio of the weight coefficients is equal to the comparative significance between the observed, that the condition is fulfilled:

$$w_k / w_{k+1} = \varphi_{k/(k+1)} \tag{17}$$

Condition 2. The final values of the weighted coefficients should satisfy the condition of mathematical transitivity

$$\varphi_{k/(k+1)} \times \varphi_{(k+1)/(k+2)} = \varphi_{k/(k+2)} \tag{18}$$

Step 4. Defining a model for determining the final values of the weighting coefficients of the evaluation criteria.

$$
\begin{aligned}
&\min \chi \\
&s.t. \\
&\left| \frac{w_{j(k)}}{w_{j(k+1)}} - \varphi_{k/(k+1)} \right| \leq \chi, \ \forall j \\
&\left| \frac{w_{j(k)}}{w_{j(k+2)}} - \varphi_{k/(k+1)} \otimes \varphi_{(k+1)/(k+2)} \right| \leq \chi, \ \forall j \\
&\sum_{j=1}^{n} w_j = 1, \\
&w_j \geq 0, \ \forall j
\end{aligned}
\tag{19}
$$

Step 5. Solving the model and obtaining the final weight of the criteria $(w_1, w_2, \ldots, w_n)^T$.

### 2.1.4. MARCOS

The Measurement Alternatives and Ranking according to a COmpromise Solution (MARCOS) method is based on defining the relationship between alternatives and reference values (ideal and anti-ideal alternatives). The MARCOS method is performed through the following steps [41]:

Step 1: Formation of an initial decision-making matrix.

Step 2: Formation of an extended initial matrix with the ideal (AI) and anti-ideal (AAI) solution:

$$
X =
\begin{array}{c}
AAI \\
A_1 \\
A_2 \\
\ldots \\
A_m \\
AI
\end{array}
\begin{bmatrix}
\begin{array}{cccc}
C_1 & C_2 & \ldots & C_n \\
x_{aa1} & x_{aa2} & \ldots & x_{aan} \\
x_{11} & x_{12} & \ldots & x_{1n} \\
x_{21} & x_{22} & \ldots & x_{2n} \\
\ldots & \ldots & \ldots & \ldots \\
x_{m1} & x_{22} & \ldots & x_{mn} \\
x_{ai1} & x_{ai2} & \ldots & x_{ain}
\end{array}
\end{bmatrix}
\tag{20}
$$

$$AAI = \min_i x_{ij} \ if \ j \in B \ and \ \max_i x_{ij} \ if \ j \in C$$

$$AI = \max_i x_{ij} \ if \ j \in B \ and \ \min_i x_{ij} \ if \ j \in C \tag{21}$$

Step 3: Normalization of the extended initial matrix ($X$):

$$n_{ij} = \frac{x_{ai}}{x_{ij}} \ if \ j \in C \tag{22}$$

$$n_{ij} = \frac{x_{ij}}{x_{ai}} \ if \ j \in B \tag{23}$$

where elements $x_{ij}$ and $x_{ai}$ represent the elements of the matrix $X$.

Step 4: Determination of the weighted matrix $V = \left[ v_{ij} \right]_{m \times n}$

$$v_{ij} = n_{ij} \times w_j \tag{24}$$

Step 5: Calculation of the utility degree of alternatives $K_i$.

$$K_i^{-} = \frac{S_i}{S_{aai}} \tag{25}$$

$$K_i^+ = \frac{S_i}{S_{ai}} \tag{26}$$

where $S_i$ ($i = 1, 2, \dots, m$) represents the sum of the elements of the weighted matrix $V$, Equation (27).

$$S_i = \sum_{i=1}^{n} v_{ij} \tag{27}$$

Step 6: Determination of the utility function of alternatives $f(K_i)$.

$$f(K_i) = \frac{K_i^+ + K_i^-}{1 + \frac{1 - f(K_i^+)}{f(K_i^+)} + \frac{1 - f(K_i^-)}{f(K_i^-)}}; \tag{28}$$

where $f(K_i^-)$ represents the utility function in relation to the anti-ideal solution, while $f(K_i^+)$ represents the utility function in relation to the ideal solution.

Utility functions in relation to the ideal and anti-ideal solution are determined by applying Equations (29) and (30):

$$f(K_i^-) = \frac{K_i^+}{K_i^+ + K_i^-} \tag{29}$$

$$f(K_i^+) = \frac{K_i^-}{K_i^+ + K_i^-} \tag{30}$$

Step 7: Ranking the alternatives is based on the final values of utility functions. It is desirable that an alternative has the highest possible value of the utility function.

### 2.2. Description of Selected Problems to Solve

2.2.1. Choosing the Masonry Wall Material

One of the more common decision problems faced by the investor or the general contractor is the choice of material for the construction of walls in residential buildings. The most frequently used solutions include, e.g.,: solid bricks, cellular concrete blocks, ceramic blocks, and silicate blocks (see Figure 3).

Each of these materials is characterized by different technical parameters and price (their values are the authors' estimations based on [84–87]). The paper presents a comparison of these four materials taking into account the following criteria: material cost, block consumption, acoustic insulation, thermal insulation, and ease of processing. A detailed list of material parameters is presented in Table 4.

**Table 4.** Examples of criteria for building materials considered by the decision-maker.

| Criteria | Solid Bricks | Cellular Concrete Blocks | Ceramic Blocks | Silicate Blocks |
|---|---|---|---|---|
| Material cost | 90.21 PLN/m² | 71 PLN/m² | 56 PLN/m² | 64 PLN/m² |
| Blocks consumption | 93 items/m² | 8.33 items/m² | 10.7 items/m² | 15 items/m² |
| Acoustic insulation | 47 dB | 38 dB | 42 dB | 47 dB |
| Thermal insulation | 3.03 W/m² K | 0.68 W/m² K | 1.31 W/m² K | 3.22 W/m² K |
| Ease of processing | 2 out of 6 | 5 out of 6 | 4 out of 6 | 3 out of 6 |

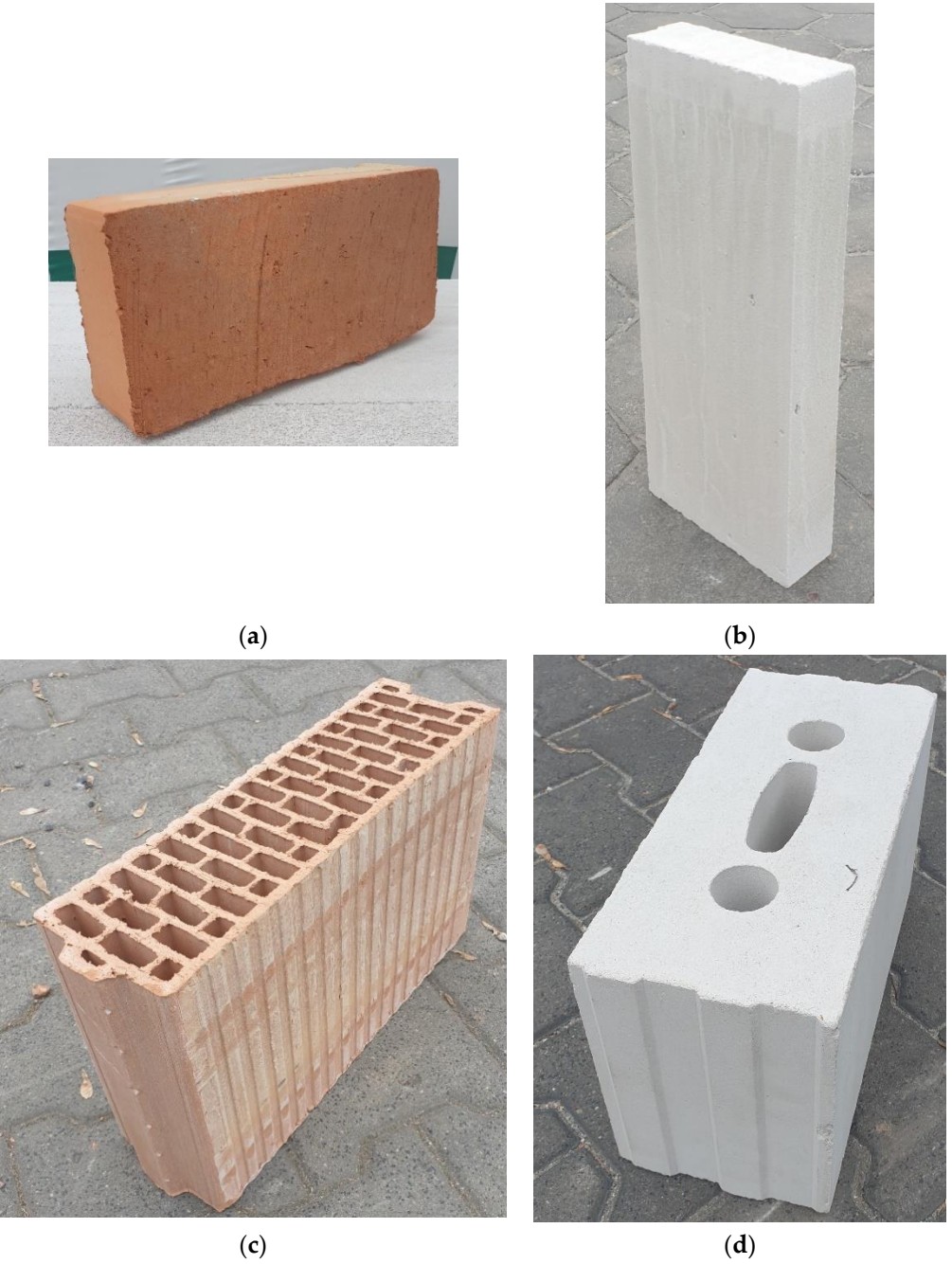

**Figure 3.** (**a**) solid bricks, (**b**) cellular concrete blocks, (**c**) ceramic blocks, (**d**) silicate blocks [own photos].

2.2.2. Choosing the Facade System

Another, inevitable decision problem for which the investor must find a solution is the choice of the elevation finishing method. The paper presents a comparison of four variants such as ETICS system, clinker cladding, natural stone, and fiber-cement panels finishing [88–91] (see Figure 4). To consider the systems mentioned above five criteria are taken into account: cost in PLN per m², execution time, aesthetics in visual reception, ease of access to the material, and durability.

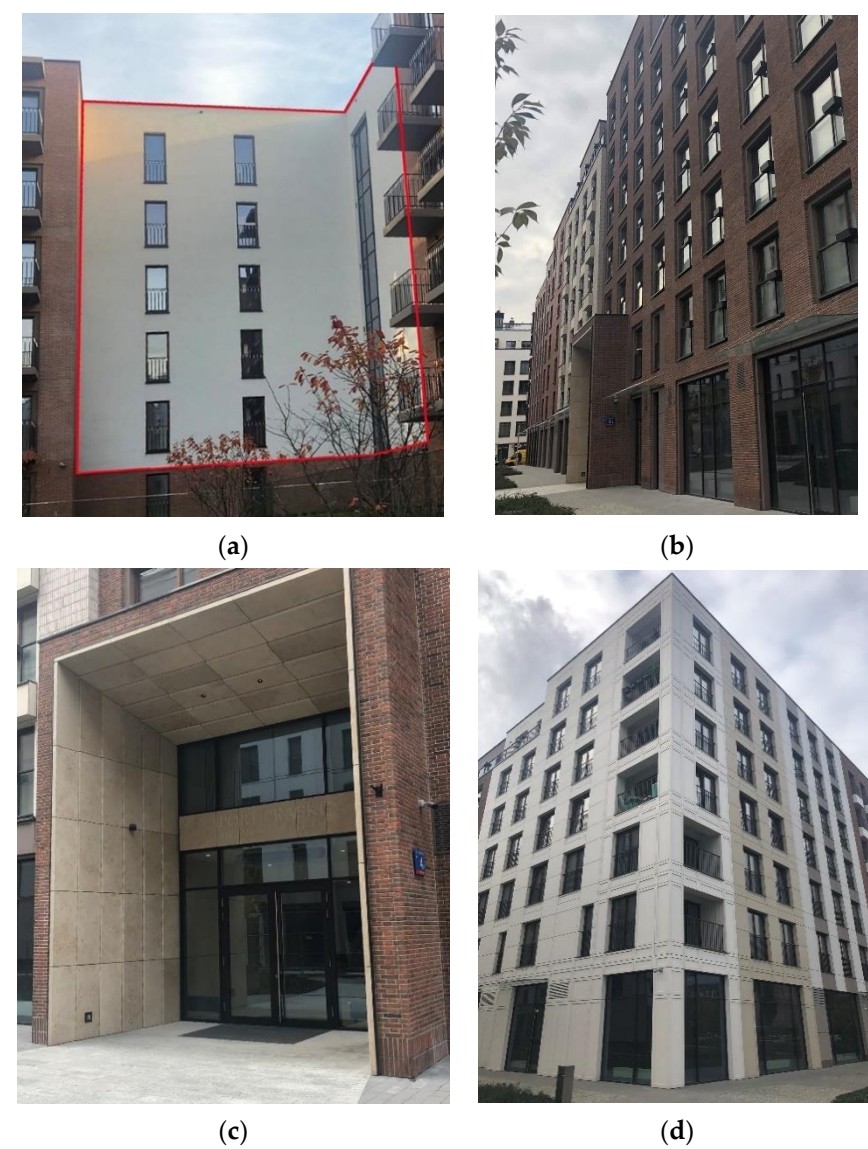

**Figure 4.** (**a**) ETICS system, (**b**) clinker cladding, (**c**) natural stone, (**d**) cement panels finishing [own photos].

A 4-point scale was used to award grades, in which 4 is the best grade and 1 is the worst grade possible. Also, the highest grade is assigned to the system with the shortest assembling time. In the case of the cost criterion, the market prices of assembling the facade systems are applied based on [27]. The rest of the ratings are given based on experts' and producers' opinions [27]. The assessment is presented in Table 5 below.

**Table 5.** Final grades awarded in terms of the criteria taken into consideration by the decision-maker.

| Criteria | ETICS | Clinker Cladding | Natural Stone Panel | Fiber-Cement Panel |
|---|---|---|---|---|
| Cost in PLN/m$^2$ (rank) | 200 (4) | 417 (3) | 656 (2) | 1435 (1) |
| Execution time | 4 out of 4 | 2 out of 4 | 2 out of 4 | 3 out of 4 |
| Visual reception | 1 out of 4 | 3 out of 4 | 4 out of 4 | 2 out of 4 |
| Ease of access | 4 out of 4 | 3 out of 4 | 1 out of 4 | 3 out of 4 |
| Durability | 1 out of 4 | 3 out of 4 | 4 out of 4 | 3 out of 4 |

## 3. Results

### 3.1. The Sequences of Solutions for Masonry Wall Materials

According to the calculations in the AHP assessment method, the following criteria weights were obtained (Figure 5).

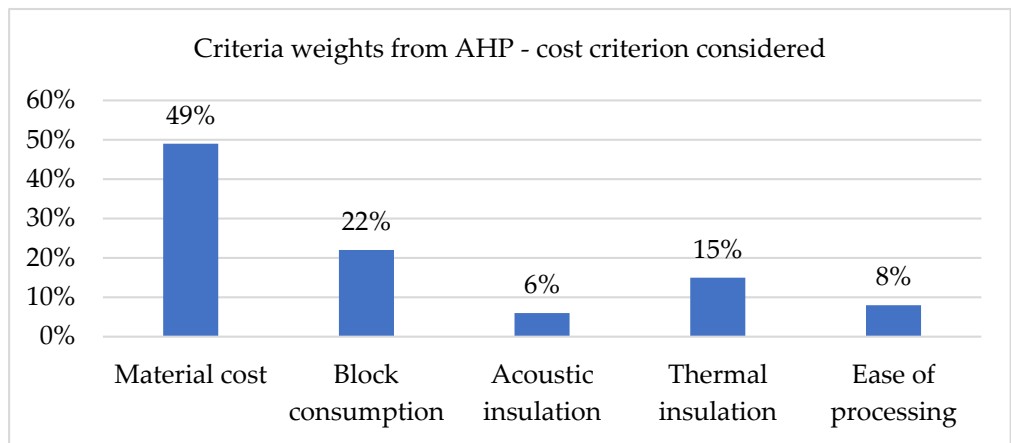

**Figure 5.** Criteria weights obtained in the AHP method in the variant of calculations taking into account the cost criterion.

After eliminating the material cost criterion and leaving the remaining data as in Table 5. As a result of such proceedings, the following results are obtained (Figure 6).

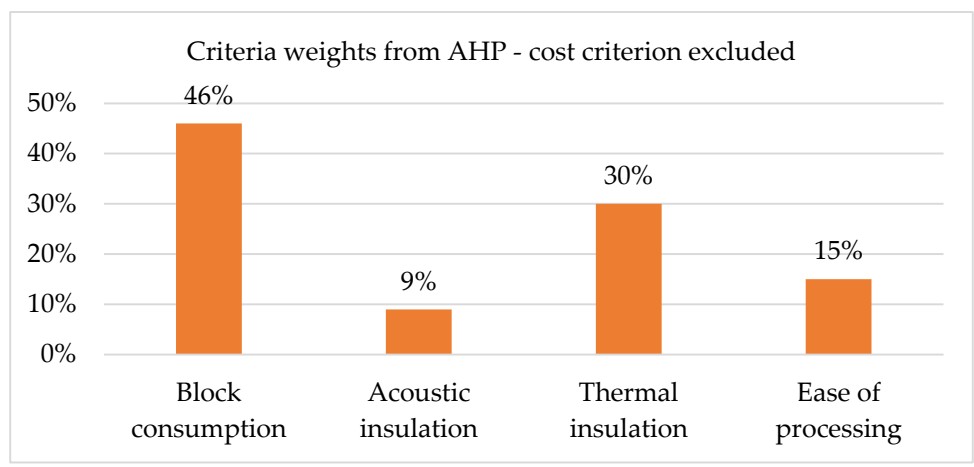

**Figure 6.** Criteria weights obtained in the AHP method in the variant of calculations without consideration of the cost criterion.

Using criteria weights from the calculations according to the AHP method, assumptions for calculations in the TOPSIS method are formulated. A 4-point gradation scale was used to prepare the X matrix. The best possible solution was assigned a score of 4, while the worst solution was marked with a score of 1. The X matrix is presented below (Table 6).

**Table 6.** The X matrix in the TOPSIS method taking into account the cost criterion.

| Criteria | Solid Bricks | Cellular Concrete Clocks | Ceramic Blocks | Silicate Blocks |
|---|---|---|---|---|
| Material cost | 1 | 2 | 4 | 3 |
| Blocks consumption | 1 | 4 | 3 | 2 |
| Acoustic insulation | 4 | 2 | 3 | 4 |
| Thermal insulation | 1 | 4 | 2 | 1 |
| Ease of processing | 1 | 4 | 3 | 2 |

A similar procedure was carried out in the TOPSIS method when the material cost criterion was not taken into account. In this case, one row was removed from the X matrix with the same number of columns. As a result of the calculations carried out with the AHP and TOPSIS methods in two variants of the criteria (with and without material costs), the following results are obtained (Table 7).

**Table 7.** Results from TOPSIS and AHP methods (rank 1 means the best).

| MCDM Method | Cost Criterion Considered | | Solid Bricks | Cellular Concrete Clocks | Ceramic Blocks | Silicate Blocks |
|---|---|---|---|---|---|---|
| AHP | Yes | Result | 0.087 | 0.290 | 0.414 | 0.210 |
| | | Rank | 4 | 2 | 1 | 3 |
| | No | Result | 0.113 | 0.462 | 0.278 | 0.148 |
| | | Rank | 4 | 1 | 2 | 3 |
| TOPSIS | Yes | Result | 0.058 | 0.501 | 0.787 | 0.544 |
| | | Rank | 4 | 3 | 1 | 2 |
| | No | Result | 0.010 | 0.921 | 0.544 | 0.260 |
| | | Rank | 4 | 1 | 2 | 3 |

If the material cost criterion is taken into account, a ceramic block is the most favorable solution, without the cost criterion cellular concrete block is the most favorable.

Solving the above-described steps of the FUCOM method we obtained the following nonlinear model for both variants. The model presented on the left includes the cost criterion, while the right without this criterion:

$$\min\chi$$

$$s.t.\begin{cases} \left|\frac{w_1}{w_2}-2.30\right|\le\chi, \left|\frac{w_2}{w_4}-1.43\right|\le\chi, \left|\frac{w_4}{w_5}-1.82\right|\le\chi, \left|\frac{w_5}{w_3}-1.33\right| \\ \left|\frac{w_1}{w_4}-3.29\right|\le\chi, \left|\frac{w_2}{w_5}-2.60\right|\le\chi, \left|\frac{w_4}{w_3}-1.33\right|\chi \\ \sum_{j=1}^{5} w_j=1, \ w_j\ge 0, \forall j \end{cases} ; \begin{cases} \left|\frac{w_1}{w_3}-1.50\right|\le\chi, \left|\frac{w_3}{w_4}-2.00\right|\le\chi, \left|\frac{w_4}{w_2}-1.67\right|\le\chi \\ \left|\frac{w_1}{w_4}-3.00\right|\le\chi, \left|\frac{w_3}{w_2}-3.34\right|\le\chi \\ \sum_{j=1}^{4} w_j=1, \ w_j\ge 0, \forall j \end{cases}$$

The results of FUCOM method are presented in Figure 7 for both variants.

Table 8 shows the results obtained using integrated FUCOM-MARCOS methods for both price criterion variants.

**Table 8.** Results from FUCOM-MARCOS methods for masonry wall materials (rank 1 means the best).

| Ai | Si | With the Cost Criterion Considered | | | | | |
|---|---|---|---|---|---|---|---|
| AAI | 0.266 | Ki− | Ki+ | F(K−) | F(K+) | F(Ki) | Rank |
| A1 | 0.297 | 1.117 | 0.297 | 0.210 | 0.790 | 0.281 | 4 |
| A2 | 0.723 | 2.723 | 0.723 | 0.210 | 0.790 | 0.685 | 2 |
| A3 | 0.836 | 3.147 | 0.836 | 0.210 | 0.790 | 0.791 | 1 |
| A4 | 0.617 | 2.322 | 0.617 | 0.210 | 0.790 | 0.584 | 3 |
| AI | 1.000 | | | | | | |
| | | Without Consideration of the Cost Criterion | | | | | |
| AAI | 0.273 | Ki− | Ki+ | F(K−) | F(K+) | F(Ki) | Rank |
| A1 | 0.319 | 1.167 | 0.318 | 0.214 | 0.786 | 0.301 | 4 |
| A2 | 0.956 | 3.500 | 0.955 | 0.214 | 0.786 | 0.902 | 1 |
| A3 | 0.675 | 2.473 | 0.674 | 0.214 | 0.786 | 0.637 | 2 |
| A4 | 0.470 | 1.723 | 0.470 | 0.214 | 0.786 | 0.444 | 3 |
| AI | 1.000 | | | | | | |

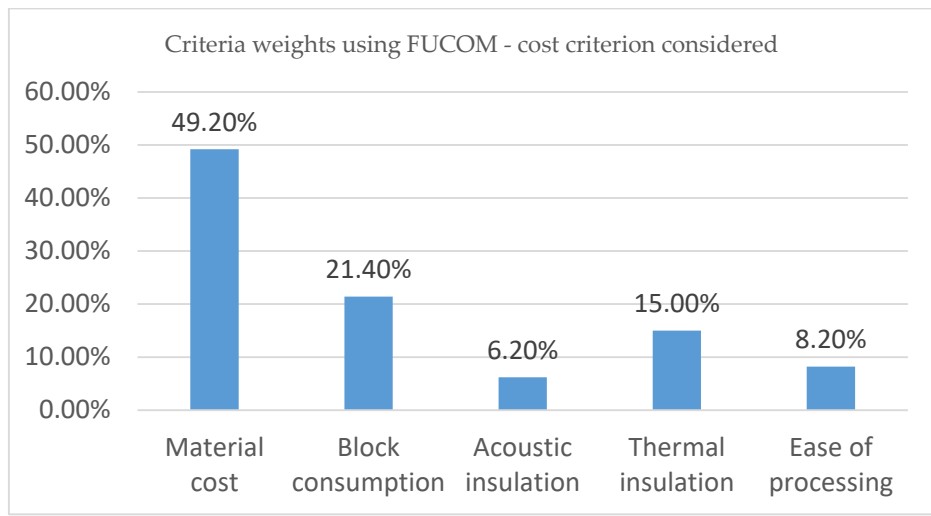

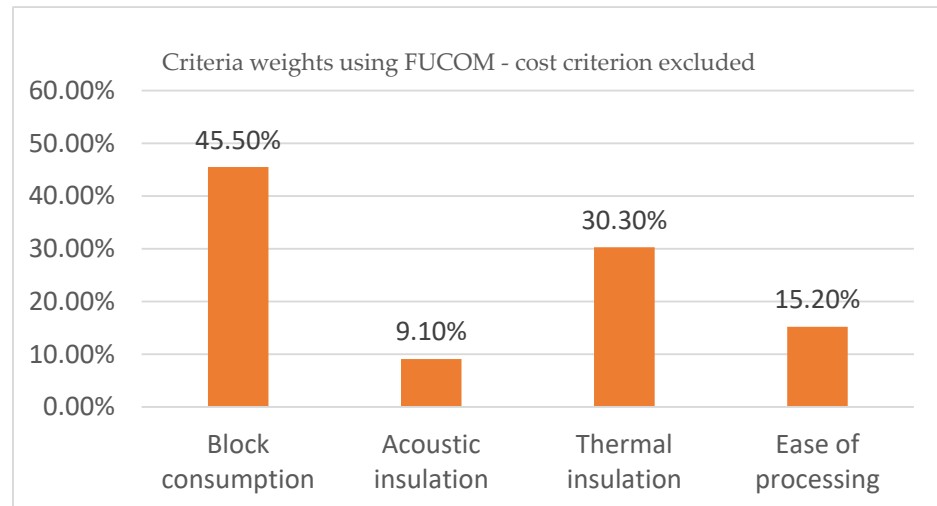

**Figure 7.** Criteria weights obtained using FUCOM method.

### 3.2. *The Sequences of Solutions for Facade System*

According to the calculations in the AHP assessment method, the following criteria weights are obtained (Figure 8).

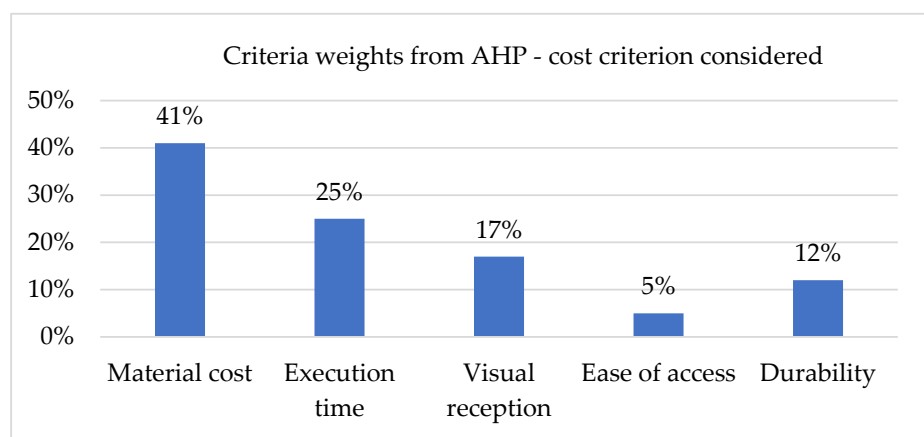

**Figure 8.** Criteria weights obtained in the AHP method in the variant of calculations taking into account the cost criterion.

After eliminating the cost criterion and leaving the remaining criteria the following results are obtained (Figure 9).

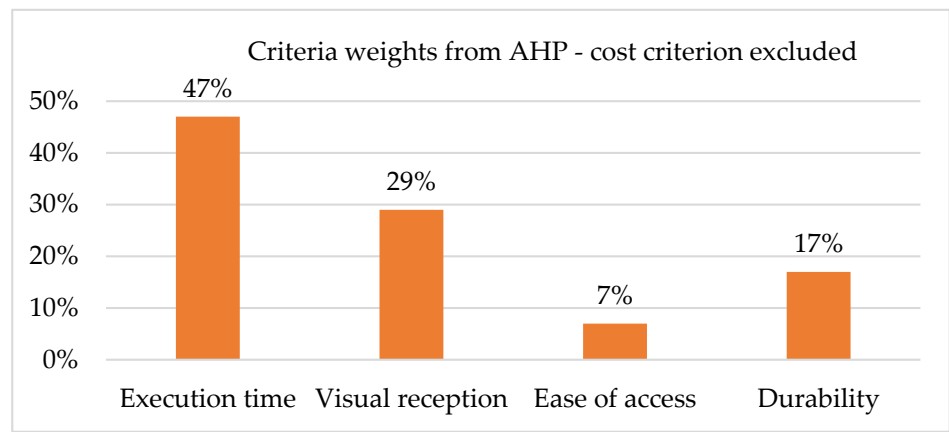

**Figure 9.** Criteria weights obtained in the AHP method in the variant of calculations without considering the cost criterion.

Using criteria weights from the calculations according to the AHP method, assumptions for calculations in the TOPSIS method are formulated. The *X* matrix is consistent with Table 4 above. As a result of the calculations carried out with the AHP and TOPSIS methods in two variants of the criteria (with and without material costs), the following results are obtained (Table 9).

**Table 9.** Results from AHP and TOPSIS obtained for the choice of the elevation finishing method.

| MCDM Method | Cost Criterion Considered | | ETICS | Clinker Cladding | Natural Stone Panel | Fiber-Cement Panel |
|---|---|---|---|---|---|---|
| AHP | Yes | Result | 3.125 | 2.750 | 2.538 | 2.008 |
| | | Rank | 1 | 2 | **3** | 4 |
| | No | Result | 2.634 | 2.529 | 2.837 | 2.716 |
| | | Rank | 3 | 4 | 1 | 2 |
| TOPSIS | Yes | Result | 0.671 | 0.574 | 0.442 | 0.243 |
| | | Rank | 1 | 2 | 3 | 4 |
| | No | Result | 0.448 | 0.446 | 0.552 | 0.481 |
| | | Rank | 3 | 4 | 1 | 2 |

If the material cost criterion is considered ETICS system is the most favorable solution. The omission of the cost criterion results in natural stone panel finishing being the most favorable.

The results of FUCOM method for this example are as follow:

$$\textit{With} \cos t: w_1 = 0.401, w_2 = 0.259, w_3 = 0.171, w_4 = 0.050, w_5 = 0.118,$$
$$\textit{Without} \cos t: w_1 = 0.469, w_2 = 0.285, w_3 = 0.072, w_4 = 0.174,$$

Table 10 shows the results obtained using integrated FUCOM and MARCOS methods for both variants in the second example.

**Table 10.** Results from FUCOM-MARCOS methods for facade system.

| Ai | Si | With the Cost Criterion Considered | | | | | | |
|---|---|---|---|---|---|---|---|---|
| **AAI** | **0.270** | **Ki−** | **Ki+** | **F(K−)** | **F(K+)** | **F(Ki)** | **Rank** | |
| A1 | 0.782 | 2.896 | 0.783 | 0.213 | 0.787 | 0.740 | 1 | |
| A2 | 0.576 | 2.133 | 0.577 | 0.213 | 0.787 | 0.545 | 2 | |
| A3 | 0.553 | 2.048 | 0.554 | 0.213 | 0.787 | 0.524 | 3 | |
| A4 | 0.462 | 1.709 | 0.462 | 0.213 | 0.787 | 0.437 | 4 | |
| AI | 0.999 | | | | | | | |
| **Ai** | **Si** | **Without Consideration of the Cost Criterion** | | | | | | |
| **AAI** | **0.367** | **Ki−** | **Ki+** | **F(K−)** | **F(K+)** | **F(Ki)** | **Rank** | |
| A1 | 0.656 | 1.786 | 0.656 | 0.269 | 0.731 | 0.597 | 3 | |
| A2 | 0.633 | 1.723 | 0.633 | 0.269 | 0.731 | 0.576 | 4 | |
| A3 | 0.712 | 1.937 | 0.712 | 0.269 | 0.731 | 0.648 | 1 | |
| A4 | 0.679 | 1.848 | 0.679 | 0.269 | 0.731 | 0.618 | 2 | |
| AI | 1.000 | | | | | | | |

*3.3. Results Summarized*

The two cases of MCDM methods applied to construction problems (the choices of masonry wall material and facade system) are calculated with three MADM methods: AHP, TOPSIS, and FUCOM-MARCOS (which combine two methods as the name presents). For these six analyses results are found in two modes: including the cost criterion in MCDM analysis or the cost criterion is analysed with results from MCDM method. All results, together with prices of variants are presented in Tables 11 and 12.

**Table 11.** Summarized results for the choice of masonry wall material.

| MCDM Method | Cost Criterion Considered | | Solid Bricks | Cellular Concrete Clocks | Ceramic Blocks | Silicate Blocks |
|---|---|---|---|---|---|---|
| AHP | Yes | Result | 0.087 | 0.290 | 0.414 | 0.210 |
| | | Rank | 4 | 2 | 1 | 3 |
| | No | Result | 0.113 | 0.462 | 0.278 | 0.148 |
| | | Rank | 4 | 1 | 2 | 3 |
| TOPSIS | Yes | Result | 0.058 | 0.501 | 0.787 | 0.544 |
| | | Rank | 4 | 3 | 1 | 2 |
| | No | Result | 0.010 | 0.921 | 0.544 | 0.260 |
| | | Rank | 4 | 1 | 2 | 3 |
| FUCOM-MARCOS | Yes | Result | 0.281 | 0.685 | 0.791 | 0.584 |
| | | Rank | 4 | 2 | 1 | 3 |
| | No | Result | 0.301 | 0.902 | 0.637 | 0.444 |
| | | Rank | 4 | 1 | 2 | 3 |
| Cost of variants PLN/m$^2$ | | | 90,21 | 71.00 | 56.00 | 64.00 |

**Table 12.** Summarized results for the choice of facade system.

| MCDM Method | Cost Criterion Considered | | ETICS | Clinker Cladding | Natural Stone Panel | Fiber-Cement Panel |
|---|---|---|---|---|---|---|
| AHP | Yes | Result | 3.125 | 2.750 | 2.538 | 2.008 |
| | | Rank | 1 | 2 | **3** | 4 |
| | No | Result | 2.634 | 2.529 | 2.837 | 2.716 |
| | | Rank | 3 | 4 | 1 | 2 |
| TOPSIS | Yes | Result | 0.671 | 0.574 | 0.442 | 0.243 |
| | | Rank | 1 | 2 | 3 | 4 |
| | No | Result | 0.448 | 0.446 | 0.552 | 0.481 |
| | | Rank | 3 | 4 | 1 | 2 |
| FUCOM-MARCOS | Yes | Result | 0.740 | 0.545 | 0.524 | 0.437 |
| | | Rank | 1 | 2 | 3 | 4 |
| | No | Result | 0.597 | 0.576 | 0.648 | 0.618 |
| | | Rank | 3 | 4 | 1 | 2 |
| Cost of variants PLN/m$^2$ | | | 200 | 417 | 656 | 1435 |

## 4. Discussion

The results from three methods, which include price criterion, indicate that ceramic blocks are the most favorable wall construction material. On the other hand, solid bricks are the least favorable. The ceramic blocks and cellular concrete blocks are ranked as 2nd or 3rd depending on the method used. Even though, the results are slightly inconsistent, for all three MCDM methods cost is the critical criterion, skewing the results (see Table 4).

Removing the price criterion from the inputs to the MCDM methods results in a differently ordered ranking. Moreover, the rankings become consistent. The most favorable are cellular concrete blocks. The sequence of the rest of the materials is the same from all methods applied: 2nd—ceramic blocks, 3rd—silicate blocks, 4th—solid bricks. When block consumption, acoustic insulation, thermal insulation, and the ease of processing are considered the cellular concrete blocks are ranked the highest, indicating to be the most suitable solution from technical properties and technological perspectives. However, the other two solutions (ceramic blocks and silicate blocks) are cheaper than cellular concrete blocks (which are consistently ranked higher). There is a logic behind it as the construction industry is highly capital consuming. The structural wall price increase of 10%, considering the area of the walls of a thousand m$^2$ may significantly influence the financial result of the decision-maker employer.

Let us present the result from the FUCOM-MARCOS method (with excluded the price criterion) together with the prices on the horizontal axis (Figure 10).

As the ranks of the masonry wall solutions are the same, the general shape of Figure 10 would be the same if it is prepared based on results from AHP and TOPSIS methods. This way of the results' presentation is of great importance. The method points to cellular concrete blocks as the best solution (according to the decision maker's preferences). Beginning the analysis from the cheapest solution, it is obvious that the choices of silicate blocks or solid bricks are irrational. They give much less of the technical utility (defined by the decision-maker) for much higher prices. The cellular concrete blocks (the highest utility) and ceramic blocks (the cheapest) trade-off is the only one that should be considered by the decision-maker.

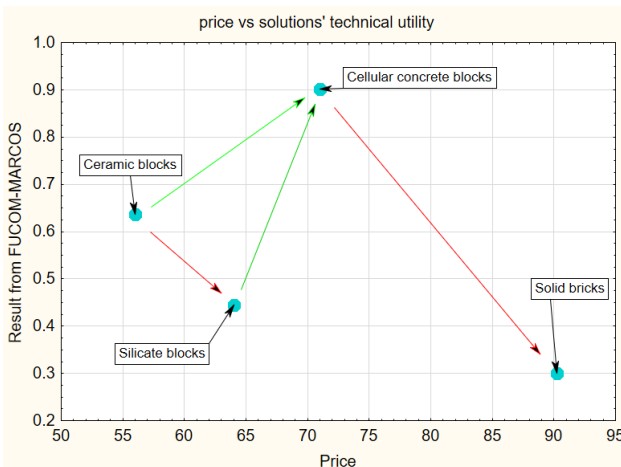

**Figure 10.** Results from the FUCOM-MARCOS method presented as a discrete function of prices (wall structural materials case).

Preparing a similar scatterplot for the case with price criterion included in the MCDM method (see Figure 11) is possible, but it is not so informative and useful (as for the case where the price criterion is not considered in the MCDM method). Beginning the analysis from the cheapest and the best solution (pointed by the method)—ceramic blocks, the change to the other (2nd or 3rd) solution can be theoretically considered. However, for cellular concrete blocks (2nd position in the ranking) and silicate blocks (3rd), the price increase can be observed together with the loss of value (calculated through the MCDM method). This is misleading because of the influence of price on the position in the ranking (on the value got from the MCDM method). Solid bricks shouldn't be considered at all according to the worst position in the ranking and the highest price.

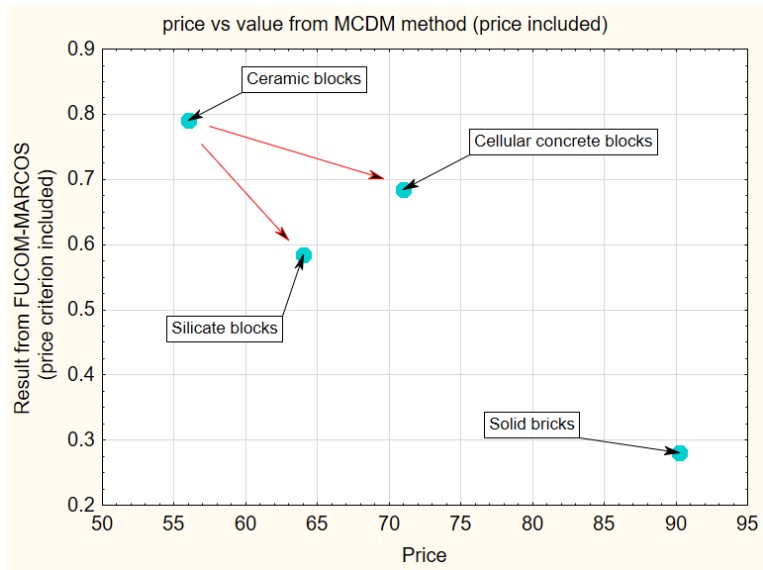

**Figure 11.** Results from FUCOM-MARCOS method with price criterion considered (wall structural materials case).

In the second case—the choice of the facade cladding system—there are also qualitative criteria as durability (assessed with Delphi method), visual reception (assessed through questionnaire form in [27]). Only the prices of the systems are the quantitative input to the MCDM methods. The results (with price criterion considered) can also be presented in the form of a scatterplot (see Figure 12). All three methods ranked four types of facade

cladding the same. When the price criterion is included, it dominates and the best system is the cheapest one (ETICS facade). The highest price is seven times higher than the lowest one. Regardless of the type of MCDM method applied, the method will be very price sensitive. The higher the price of the cladding system the lower the rank is assessed.

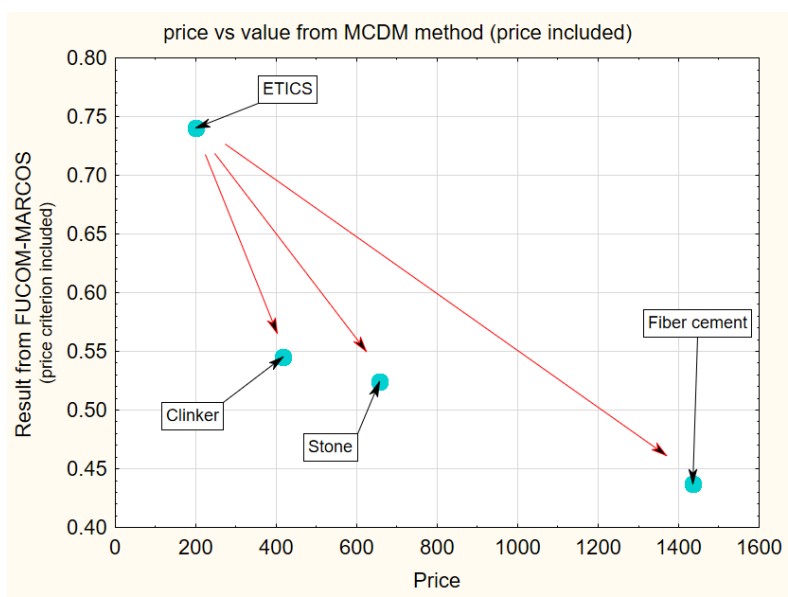

**Figure 12.** Results from FUCOM-MARCOS method with price criterion considered (facade cladding case).

Figure 12 indicates that the more is paid for the facade system, the lower is its value for the decision-maker. This statement is questionable. Figure 13 presents the results from the FUCOM-MARCOS method, where the price criterion is not considered.

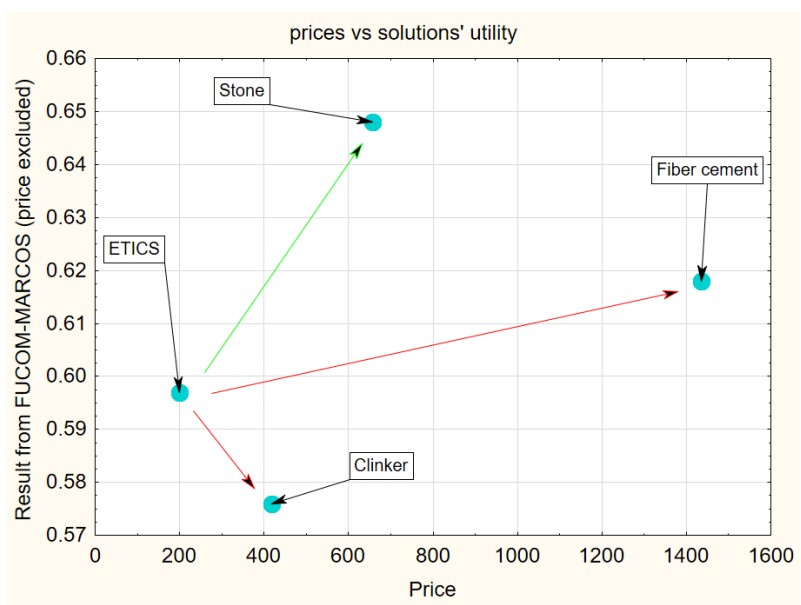

**Figure 13.** Results from the FUCOM-MARCOS method presented as a discrete function of prices (facade cladding case).

Considering the cheapest solution (ETICS facade system), clinker cladding should be excluded, as its properties—according to the decision-maker preferences—are lower than ETICS, but the price is higher. The fiber-cement cladding provides insignificant gains in

the value (defined by the decision-maker), but its price is 7 times higher than the ETICS price. It should be excluded from reasonable solutions. The only alternative to the ETICS is the stone facade. It is over three times more expensive (not seven times as fiber-cement panels), but provide the highest increase of the value recognized by the decision-maker. The AHP and TOPIS methods produced the same sequence of facade systems indicating the same results as presented in Figure 13.

There is an extremely important conclusion, which was so far implicit: if the cost criterion is applied in MCDM analysis, it means that economic issues are important for the decision-maker. The higher the cost of the variant, the lower the decision maker's willingness to choose the variant. The cost increase may lower the efficiency or capabilities of the entity, the decision-maker acts for. This is the very important reason, to exclude the price criterion from MCDM methods, and to analyse the results achieved through them at the final stage considering the costs of variants. This procedure—described below—can be named MCDM-CCAF (multi-criteria decision making with cost criterion analysed at the final stage). It is proposed and strongly recommended to apply it for MCDM methods—where costs are one of the factors influencing the result—proceeding with the following procedure:

Stage 1: exclude the cost criterion from the set of criteria and calculate the ranking of variants with the MCDM method.

Stage 2: order the variants in cost ascending sequence, then prepare a scatterplot (Figures 10 and 13 as examples).

Stage 3: start the analysis from the cheapest variant and exclude the variants of the lower value (from the MCDM method) than the cheapest.

Stage 4: make a decision based on the trade-off: choose the cheapest variant (it stops the procedure), or consider the variant providing higher value, but which is just a step more expensive.

Stage 5: if the more expensive variant is chosen, exclude from the analysis all variants providing the value lower than the variant being just analysed.

Stage 6: make a decision based on the trade-off: choose the presently analysed variant (it stops the procedure), or consider the variant providing higher value, but which is just a step more expensive (if exist; if not the procedure is stopped). Go to Stage 5.

This 6-stages procedure (CCAF) has the following easy noticeable advantages:

- It protects the decision-maker from choosing the variant which has a lower value, but its cost is higher
- Every next variant, if considered, brings an increase of the value
- Economic analysis (based on the cost of the analysed variant) can be made on each intermediate stage (not only on the final one)
- As a result of these economic analyses, the decision-maker can stop the procedure at every analysed variant, i.e., he can choose the variant providing fair technical properties paying a fair price for it. The choice of technically the best solution is not compulsory.

For a clear explanation another example is prepared (see Table 13).

**Table 13.** Data to the example. Stages 1 and 2 of the proposed MCDM-CCAF procedure.

| Label of Variant | A | B | C | D | E | F |
|---|---|---|---|---|---|---|
| Cost of variant | 100 | 120 | 125 | 140 | 150 | 155 |
| Values from MCDM method (cost excluded) | 0.36 | 0.52 | 0.27 | 0.88 | 0.92 | 0.69 |
| Ranking (based on values) | 5 | 4 | 6 | 2 | 1 | 3 |

The proposed 6-stages CCAF procedure is illustrated in Figure 14.



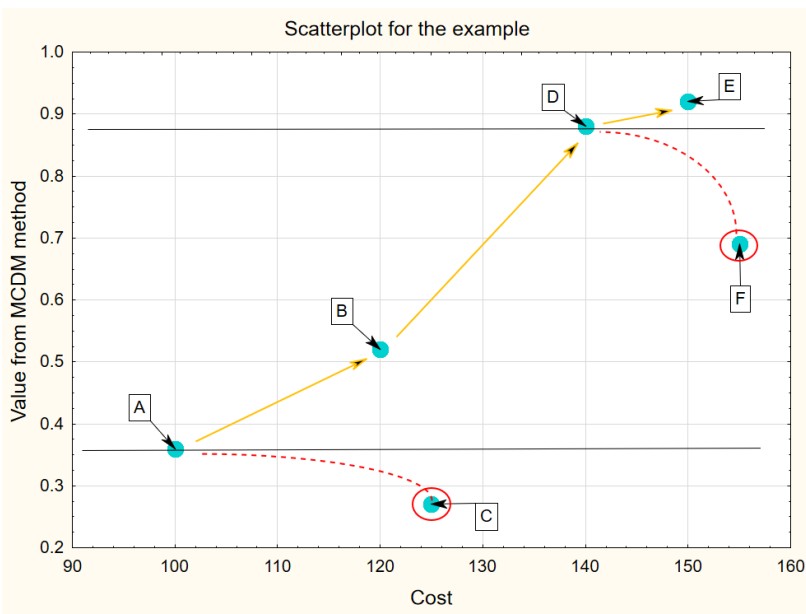

**Figure 14.** The sequential analysis of the results from the MCDM method—Stages 3 to 6.

Initially, the cheapest variant (A) is chosen. As the variant C is assessed lower, it is excluded from the available variants. The closest step up with the price is now the variant B. Choosing If it is chosen (instead of A) as a variant bringing more of the value than A, A is excluded from the available variants. Except for variant B, there are now also D, E, F available. Then the choice has to be done: staying with B (according to economic reasons) or considering D as more valuable and more expensive (just one step more expensive than B; for the same reason the choices of F or E are forbidden). If D is chosen, all less valuable variants (B and F) should be excluded from the set of available variants. Finally, the choice between D and E should be made. It is up to the decision-maker (depends on his price perception) which variant will be chosen: of the lower value and cheaper (D) or the variant providing the highest value and more expensive than D (i.e., E). The 6-stage CCAF procedure protects from choosing unreasonable variants C and F, as well as, allow considering economic issues while deciding if to choose the more valuable variants. It is not possible to present the cases of all possible mutual location of variants on the scatterplot, however, they were analysed. Applying the MCDM-CCAF procedure, the set of possible choices are Pareto-optimal i.e., for each chosen price the prosed variant is of the best value (value in terms of decision-maker preferences). The proposed MCDM-CAFF procedure can be presented as a flowchart (Figure 15).

In the traditional approach, where the cost criterion serves as an input to the MCDM method (as in the 37 articles mentioned before), this kind of analysis is not possible. The technical, technological, and more intangible (as visual reception) preferences of the decision-maker are mixed with the economic preferences (accepted level of cost to be spent on a certain variant).

Other advantages of the proposed 6-stage method are:

- The producers usually price their products based on the properties of the products, but also based on the competitors' prices, the state of the market, etc. Accepting the price into MCDM analysis means the influence of the market issues on the prepared assessment. Considering the price after MCDM analysis (i.e., applying proposed CCAF), makes the choices more explainable. Features of the product are transformed into the ranking based on the decision maker's preferences (through the MCDM method), and then, the decision-maker's price sensitivity is matched to the market price of the products (stages 3 to 6).

- In the traditional MCDM, the variant assessed as the 2nd (or with lower rank) is presented as a worse one (also because of its price). In economic decisions (as the decision based on cost is) the ratio: value (received) to cost (spent on it), is very important. The proposed method is based on this concept
- All advantages of MCDM methods are kept, as MCDM is a part of the 6-stage method (jointly named as MCDM-CCAF).
- Price changes (caused by market processes or achieved in negotiations) can be easily adjusted in the scatterplot (without the necessity of repeating MCDM analysis, as other than price criteria e.g., sound insulation or number of block per m$^2$ are stable).

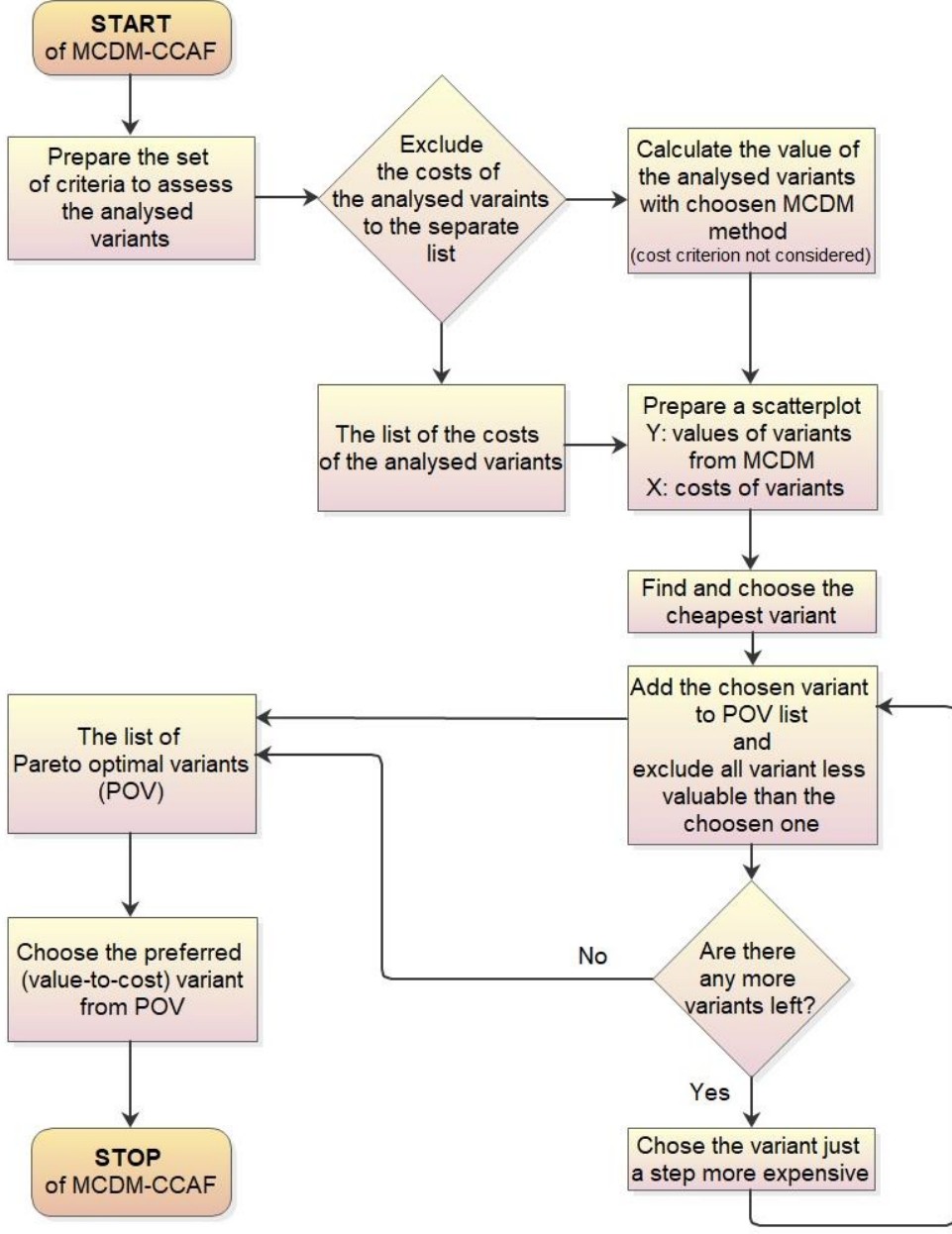

**Figure 15.** The flow-chart of MCDM-CCAF decision-making process.

Disadvantages of MCDM-CCAF:
- Higher analytical effort (if compared to a solely applied, pure MCDM method).
- Only one cost criterion can be considered (if there are more, they should be combined into one).

## 5. Conclusions

Among the analysed articles there are 79 where multi-criteria decision-making methods are applied to construction industry problems. In nearly 50% of them (i.e., in 37) the costs of analysed variants serve as an input to MCDM methods. Considering cost as a criterion in evaluating the decision-maker's preferences makes the result from the MCDM method cost-sensitive. The influence of technical, technological, and other—"intangible"—criteria is analysed then together with the influence of cost on the final ranking. In all these 37 analysed articles the cost criterion is considered. It is proposed to exclude the cost from the set of criteria (i.e., from the MCDM analysis input) and consider it when the result from the MCDM method is already calculated. Then, with the use of the proposed procedure (named MCDM-CCAF) variants are filtered. Only Pareto-optimal variants are left to the decision-maker, to choose. Based on the budget, the decision-maker chooses the acceptable price. There is only one variant assigned to each price, providing the maximum value (the value other than cost-based value, calculated through the MCDM method, according to the decision-maker preferences). Moreover, the process of choosing the variant (from the set of Pareto-optimal variants) is realized in the CCAF procedure step by step (see Stages 3 to 6 described in the discussion section). Each step shifts the focus to consider just one step up (in terms of cost) and allows the decision-maker to decide (at each price level) which variant is the most suitable.

It is strongly recommended to apply the proposed MCDM-CCAF method to the problems where the costs of the variants are important. There are numerous advantages of excluding the cost criterion from pure MCDM analysis. The most important advantages of the MCDM-CCAF method are as follows: versatility (it can be applied with any MCDM method), economic sensitivity (each Pareto-optimal variant provide the highest value, for the price chosen), easiness of application (finding the result from MCDM-CAFF is not more complicated than the complexity of applied MCDM method), time-saving (in case of price changes).

**Author Contributions:** Conceptualization, H.A.; methodology, H.A., A.N., Ž.S., and M.G.; validation, H.A., Ž.S., and K.S.; formal analysis, H.A., and K.S.; investigation, A.N., and M.G.; resources, A.N., and M.G.; data curation, A.N., and M.G.; writing—original draft preparation, H.A., A.N., Ž.S., and M.G.; writing—review and editing, H.A., Ž.S., and K.S.; visualization, H.A., A.N., Ž.S., and M.G.; supervision, H.A.; project administration, A.N., and H.A.; funding acquisition, A.N., K.S., H.A., and Ž.S. All authors have read and agreed to the published version of the manuscript.

**Funding:** This research is partly financed by Warsaw University of Technology "Research grant of Scientific Council of the Discipline of Civil Engineering and Transport".

**Institutional Review Board Statement:** Not applicable.

**Informed Consent Statement:** Not applicable.

**Data Availability Statement:** Input data are collected by M.G. They are published in [27] as a master thesis. They are available (in Polish) on request sent to corresponding author.

**Conflicts of Interest:** The authors declare no conflict of interest.

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
