# Peer review of "Pareto Optimal Decisions in Multi-Criteria Decision Making Explained with Construction Cost Cases"

_symmetry, doi:10.3390/sym13010046_

Round 1
Reviewer 1 Report
The article is quite interesting. Many literature sources have been examined. Therefore, we can say that the authors are well acquainted with the subject matter of their article.
In my opinion, there are some inaccuracies and errors in the article:
- Line 130 – unclear statement.
- In formula (1), it is not clear in the denominator whether it is a comma or some other sign (Line 134).
- Formula (2) contains two commas (Line 135).
- There are no punctuation marks after formulas 4, 6-9, 12-15, 16-30.
- Illogically formulated model (19): the objective function is given as min c, and under constraining conditions the same c (=c).
- Figure 4 must contain markings a), b), c), d) - line 268.
- The names of the graphs in Figure 5 (Line 282) and in Figure 6 (Line 288) need to be adjusted; as well as in the Figure 8 (Line 323) and in Figure 6 (Line 328).
- Unclear calculations on line 310 need to be corrected.
- Line 443 – error If.
Author Response
Dear Reviewer,
Thank you for your kind assessment and careful reviewing. All your remarks have been addressed.
It is also our feeling that considering them the article is more clear.
Thank you.
Faithfully yours
Team of authors
Reviewer 2 Report
The authors obtained interesting results. The paper substantiates the necessity and effectiveness of eliminating the cost factor when making a multi-criteria decision. The research contains scientific novelty.
I suggest the following improvements.
- Well-known multi-criteria decision-making methods are described unnecessarily detailed. In particular, this refers to the description of the AHP and even TOPSIS. I suggest shortening the well-known description of these methods in Sections 2.1.1 and 2.1.2. A general description of the essence of these methods with references to known publications will suffice.
- On the other hand, the general structure of the research and the peculiarities of the combination of the methods used are presented in insufficient detail. I suggest illustrating the developed technique in the form of a flowchart.
Author Response
Dear Reviewer,
Thank you for the kind assessment. The article is enriched now by a flow-chart of the proposed MCDM-CCAF method. It is also our feeling that it made our findings more clear.
We agree that presenting AHP and TOPSIS methods enlarged the article (not bringing any new findings in this section). It is our joint decision to leave them for the three following reasons:
- It illustrates the traditional approach better (than reference and results only)
- Any MCDM method can be a part of MCDM-CCAF, so the calculations are presented in full (for the whole MCDM-CCAF)
- The readers less experienced with multi-criteria decision making methods will be provided with more comprehensive information.
Considering these, let us leave the descriptions as they are presented.
Thank you so much for all your suggestions.
Faithfully yours
Team of authors